# Metabolomics as a Potential Chemotaxonomical Tool: Application on the Selected *Euphorbia* Species Growing Wild in Serbia

**DOI:** 10.3390/plants12020262

**Published:** 2023-01-06

**Authors:** Ivana Sofrenić, Boban Anđelković, Dejan Gođevac, Stefan Ivanović, Katarina Simić, Jovana Ljujić, Vele Tešević, Slobodan Milosavljević

**Affiliations:** 1Faculty of Chemistry, University of Belgrade, Studentski Trg 12-16, 11000 Belgrade, Serbia; 2Institute of Chemistry, Technology and Metallurgy, National Institute of the Republic of Serbia, University of Belgrade, Njegoševa 12, 11000 Belgrade, Serbia; 3Serbian Academy of Sciences and Arts, Knez Mihajlova 35, 11000 Belgrade, Serbia

**Keywords:** *Euphorbia*, NMR metabolomics, PCA, OPLS-DA, biomarkers, chemotaxonomy, diterpenes, triterpenes, flavonoids

## Abstract

Chemotaxonomy presents various challenges that need to be overcome in order to obtain valid and reliable results. Individual genetic and environmental variations can give a false picture and lead to wrong conclusions. Applying a holistic approach, based on multivariate data analysis, these challenges can be overcome. Thus, a metabolomics approach has to be optimized depending on the subject of research. We used ^1^H NMR-based metabolomics as a potential chemotaxonomic tool on the selected *Euphorbia* species growing wild in Serbia. Principal components analysis (PCA), soft independent modeling by class analogy (SIMCA) and Orthogonal Projections to Latent Structures Discriminant Analysis (OPLS-DA) were used to analyze obtained NMR data in order to reveal chemotaxonomic biomarkers. The standard protocol for plant metabolomics was optimized aiming to extract more specific metabolites, which are characteristic for the *Euphorbia* genus. The obtained models were validated, which revealed that variables unique for each species were associated with certain classes of molecules according to literature data. In *E. salicifolia*, acacetin-7-*O*-glycoside (not found before in the species) was detected, and the structure of the aglycone part was solved based on 2D NMR data. In the presented paper, we have shown that metabolomics can be successfully used in *Euphorbia* chemotaxonomy.

## 1. Introduction

Metabolomics analysis has been applied in numerous areas of research related to plant biology and chemistry. Its role could be defined as the extension of traditional phytochemical investigation following the identification of biomarkers that can be statistically correlated to bioactivity changes but also as an attempt to understand complex plant mechanisms as a part of systems biology [1]. In several studies, metabolomics has been helpful in the fingerprinting of species, genotypes or ecotypes for taxonomic or biochemical (gene discovery) purposes [2,3,4,5,6,7,8,9]. For example, the PCA analysis of ^1^H NMR metabolite fingerprinting was used to discriminate five *Verbascum* species. Among these species, *V. xanthophoeniceum* and *V. nigrum* have higher amounts of the pharmaceutically important harpagoside and verbascoside, forsythoside B and leucosceptoside B [10]. Hierarchical analysis of NMR data was well correlated to the phylogenetic data [11], showing that metabolomics data can be used to make a link between chemotaxonomy and phylogeny. Great care must be taken in such analyses to avoid the influence of environmental and genetic variations of the investigated plants. An unbiased LC-MS-based metabolomics approach applied to *Lonicera* species flower buds delivered models for classification correlated to taxonomy based on morphological characteristics. Several potential biomarkers were identified using MS-MS analysis and data interpretation [12].

The Euphorbiaceae family, known as the spurge family, includes 322 genera and 8910 species. It is one of the most complex, large and diverse families of angiosperms ranging from large woody trees to simple weeds [13]. Genus *Euphorbia* is the second largest genus of flowering plants [14] with over 2100 species occurring in all temperate and tropical regions [15]. The taxonomy of *Euphorbia* is extremely difficult due to the species richness accompanied by a cosmopolitan distribution, the extreme morphological plasticity among certain species and the convergent evolution of certain morphological characters. After recent molecular studies, *Euphorbia* has been divided into four monophyletic subgenera, including Athymalus (ca. 150 species), Chamaesyce (ca. 600 species), Esula (ca. 500 species) and *Euphorbia* (ca. 800 species). Many members of the genus *Euphorbia* contain a poisonous milky-latex sap. In studies performed by Se Jin Park, the metabolomics study of 18 *Euphorbia* species from the growth chambers was carried out [16]. The latex extraction protocol for LC/MS analysis was optimized. In all tested protocols, polar solvents suitable for LC/MS analysis were used and the numbers of detected peaks were the main elimination criteria. In this study, the author indicated differences in the chemical composition of latex extracts but without further explanation and metabolites identification. In a study performed by S. El-Hawery et al., fifteen *Euphorbia* species were the subject of metabolomics profiling and searching for species which have the most biologically active compounds against human hepatoma (HepG2) and human breast adenocarcinoma (MCF-7) cell lines [17]. By exploring LC-HRMS and PCA analysis, they identify *E. lactea* and its two constituents with the molecular formulas C_16_H_18_O_8_ and C_20_H_30_O_10_, which were responsible for cytotoxic activity against MCF-7 and HepG2 cell lines. L. F. Salomé-Abarca et al. used ^1^H NMR and HPTLC metabolomics in revealing the geographical and inter species variations of two *Euphorbia* species collected in Serbia [18]. They used methanol extracts of different plant organs (leaves and roots) for their study. Their results showed that the metabolic variation of latexes within species was much more limited than between species and different organs. On the other hand, in order to monitor changes in *E. palustris* latex after fungal infection, G. Krstić et al. used the ^1^H NMR spectral data of latex CDCl_3_ extracts to investigate the effect of plant fungal infections on the chemical compositions of specific metabolites in *Euphorbia palustris* latex. The infected plants had a greater content of antifungal diterpene metabolites then plants without fungal infection [19]. These results were obtained using multivariate data analysis and in vivo experiments on fungi isolated from infected plants.

In the last three decades, species of the *Euphorbia* genus have been the subject of extensive phytochemical research [20,21,22,23]. From the results of the aforementioned studies, it is clear that the latex of this genus is rich in specific metabolites of diterpene and triterpene types. These molecules are mostly medium polar or non-polar and the application of the standard protocol for metabolomics analysis of plants is not suitable. Chemotaxonomic differences in the chemical composition of metabolites of *Euphorbia* species were typically performed by utilizing LC/MS instrumental techniques using an ESI ion source. This ionization technique is suitable for the analysis of polar and some medium polar compounds but not for mostly non-polar, such as diterpenes and triterpenes reported in the latex of the species from the *Euphorbia* genus. Furthermore, these metabolites usually remain captured in the plant material due to inefficient extraction. MS-based techniques can be useful in the identification of metabolites structure by providing molecular formula or MS/MS fragmentation fingerprints, but the NMR is required for structure identification especially for new compounds [24]. Although the biggest disadvantage of NMR is the low sensitivity compared to MS, realistic molar ratios and the non-selectivity of NMR technique largely overcome this problem, especially in cases where the sample contains compounds of a wide range of polarity as is the case with *Euphorbia* species.

Consequently, it can be assumed that the application of multivariate analysis would be suitable for a chemotaxonomic study of *Euphorbia* species, but an optimization and standardization of the extraction protocol is needed. In this paper, we propose an optimized extraction protocol for this purpose and report ^1^H NMR based metabolomics studies of some *Euphorbia* species growing wild in Serbia as a potential chemotaxonomic tool.

## 2. Results and Discussion

### 2.1. Optimization of the Extraction for the NMR-Based Metabolomic Analysis of Euphorbia Species

In order to obtain as comprehensive a picture as possible of the metabolites present in *Euphorbia* plants, we tested several solvents for their extraction. For the optimization, we used a standard protocol described by H. K. Kim et al. [25], changing solvents and running ^1^H NMR spectra with water signals suppression (Figure 1). Solvents for the extraction were selected according to the known literature data. In order to extend the polarity range as much as possible, the following combinations of solvents were tested: 1:1 mixture of deuterated methanol and potassium phosphate buffer in deuterated water (MeOD:KP-D_2_O); deuterated methanol (MeOD); 1:1 mixture of deuterated methanol and deuterated chloroform (MeOD:CDCl_3_); and deuterated chloroform (CDCl_3_). As a model system for these studies, we randomly used finely ground freeze-dried aerial parts (containing stems, leaves and flowers) of *E. salicifolia.* The criteria for the best solvent for the extraction were the number of the signals in spectra, their intensity and resolution. Additionally, different regions of spectra were integrated, and areas were normalized using the same scale for calibration in comparison to proton signals at oxygenated carbons (from δ_H_ 2.9 to 4.8) excluding solvents, in MeOD:KP-D_2_O extracts (Appendix A). In CDCl_3_ extracts, dominant signals were those of waxes, fatty acids, di- and triterpenes in the region from δ_H_ 0.2 to 1.8. More polar metabolites such as sugars and aminoamides were rather sparse, and thus, they were excluded from further investigation. The MeOD:KP-D_2_O extracts contained a large variety of metabolites, with sugar and amino acid signals dominating (in the region from δ_H_ 0.8 to 4.7). They also contained a modest amount and intensity of proton signals from *sp^2^* hybridized carbons as well as protons from *sp^3^* hybridized carbons from di- and tri terpenes skeletons and other non-polar molecules. The “sugar” spectral area contained the most intense signals, whereas those in the aromatic region have a ca. five times smaller area. Comparing the spectra obtained after MeOD and MeOD:CDCl_3_ extraction showed that more intense and clearly defined signals are observed below δ_H_ 0.8 in the latter, and the ratio of areas between the sugar and aromatic areas became much more uniform. According to these data, we had chosen 1:1 mixture of MeOD:CDCl_3_ as a solvent for this metabolomics study.

### 2.2. Multivariate Data Analysis

Principal components analysis (PCA) was performed on the NMR metabolomics fingerprints of 60 samples originating from six distinct *Euphorbia* species. Since PCA is a technique for pattern recognition and unsupervised variable reduction, a smaller number of new variables were formed containing the majority of the original variables’ variation. This analysis yielded a model with eleven principle components that explains 96% of the total variance in the data. *E. maculata* and *E. salicifolia* were clearly distinguished from the remaining samples based on the PCA score plots of the first two principal components (Figure 2a). Samples of *E. panonica* and *E. cyparissias* were separated along the third component (Figure 2b), whereas *E. amygdaloides* was separated along the fourth component (Figure 2c). There were no outliers found in any of the score plots.

The soft independent modeling by class analogy (SIMCA) model was used to confirm the difference between the analyzed *Euphorbia* species. This is a supervised pattern recognition algorithm based on the PCA of each class individually. The data were separated into training and prediction sets. The logarithmic averaged distances between each class model (DModX) were measured, and class membership was determined by comparing DModX to the critical distance (DCrit). As depicted in Figure 3a–f, all of the samples in the prediction dataset were properly identified, indicating that the model achieved 100% sensitivity and specificity.

Six OPLS-DA models were used in order to identify metabolites unique to each *Euphorbia* species investigated. Samples belonging to the species for which distinctive metabolites are to be identified are defined as belonging to one class, while the remaining samples from all other species are defined as belonging to another class. Using this method, novel variables will account for the greatest possible separation between two previously defined classes. Since the systematic variation of variables in the orthogonal model is divided into two components, one of which is linearly related to the class information and the other is orthogonal to it, model interpretation is facilitated [26]. Therefore, OPLS-DA is appropriate for identifying variables with the highest discriminatory power between two preset groups. Cross-validation, permutation testing, and CV-ANOVA were utilized to evaluate the model’s quality (see Appendix A). The most influential variables were chosen based on their impact on the projection scores of the predictive components (VIPpred). Variables having a VIPpred score greater than 1.4 deemed crucial for the separation are shown in Figure 4, Figure 5, Figure 6, Figure 7, Figure 8 and Figure 9.

The variables that were characteristic and unique for *E. seguieriana* (Figure 4) in the region from δ_H_ 7.3 to 8.8 originate from benzoate substituents of the myrsinol diterpenes type skeleton previously reported by F. Jeske et al. [27]. This assumption is further substantiated with a variable at δ_H_ 4.05 from the tetrahydrofuran ring and those in the regions Δδ_H_ = 4.8–4.95 (exomethylene double bonds), Δδ_H_ = 5.85–6.3 (other olefinic protons), acetate esters at δ_H_ ca. 2.0 and methylenes at δ_H_ 1.8 characteristic for myrsinol diretpenes isolated from *E. seguieriana* (see Appendix A). The same biomarkers were recognized by A. I. Elshamy et al., using agglomerative hierarchical clustering of 32 *Euphorbia* species based on literature data [28].

According to the literature data, diterpene polyesters and bishomoditerpene lactones are present in *E. salicifolia*. The three euphosalicins as well as a salicinolide were identified in *E. salicifolia* by Hohmann et al. [29]. Nevertheless, the highest VIP predictive values exhibited variables at δ_H_ 7.94, 7.92, 7.10, 7.08, 6.74, 6.72, 6.65, 6.54, 5.00 and 3. 91 (Figure 5). After further investigation of 2D NMR spectra (see Appendix A), it was confirmed that these signals belong to acacetin-7-*O*-glycoside (Table 1), which can be considered a chemotaxonomic marker for *E. salicifolia*. Apigenin was previously reported in the roots of *E. salicifolia* by College et al. [30], but for the first time, apigenin derivative was detected in this species. Other variables from the loading plots were correlated to signals originating from the diterpene skeleton protons, as detected in this species [27].

The loading plot of *E. amigdaloides* contains variables spread across the spectrum (Figure 6) originating from the specific jatrophane diterpenes named amygdaloidins. They contained nicotinate and angeloyl moieties as well as olefinic protons resonating in the region from δ_H_ 6.2 to 9.6 (see Appendix A). In addition, variables in the area from δ_H_ 3.0 to 4.0 are recognized as biomarkers and originate from protons at the oxygenated carbons from the jatrophane skeletons [31,32].

The *E. panonnica* also known as *E. glareosa* and *E. nicaeensis* is rich in diterpenes of the jatrophane and tigliane type [23]. These skeletal types were characterized by a five-membered ring condensed with a twelve-membered ring or seven-membered and six-membered fused rings containing different substituents belonging to jatrophane and tigliane series, respectively. A variable, which corresponds to the chemical shifts the nodal protons of the five-membered ring of tigliane type skeletons (Figure 7), was recognized as a potential biomarker for *E. panonnica* (see Appendix A). These derivatives mostly contain benzoates as aromatic substituents in comparison to jatrophans containing nicotinates. Variables that are not recognized as significant for biomarkers, but are certainly present in *E. panonica*, are proton signals from the cyclopropane ring which further supports the assumption that benzoate-substituted tiglianes could be used as chemotaxonomic markers for *E. panonica*. 

In contrast to the previously mentioned *Euphorbia* species, where the highest values of VIP-predictive scores were attributed to signals exhibiting chemical shifts of protons from different diterpene skeletons, in the loading plot of *E. cyparissias*, the most significant are the variables belonging to the triterpene signals (Figure 8). Previously mentioned protons from the cyclopropane ring (δ_H_ 0.41) together with methyl groups of triterpenes (from δ_H_ 0.93 to 1.1), methylene skeleton signals at δ_H_ 1.72 and 1.92–1.95 as well as a broad singlet signal at δ_H_ 4.57 were recognizable in the spectra of acetylated cycloartane [18]. Additionally, two signals in the aromatic region at δ_H_ 8.11 and 8.06 are characteristic for *E. cyparissias* and responsible for the separation (see Appendix A).

Accordant to the literature data, *E. maculata* is rich in the triterpene type of special metabolites [33] and polyphenols [34]. The variables from δ_H_ 0.75 to 1.07 and 1.66 characteristic for the *E. maculata* (Figure 9) match to the methyl groups signals from triterpene skeletons. Variables from δ_H_ 3.0 to 4.5 were from proton bonded to oxygenated carbons, and those from δ_H_ 5.1 to 7.05 belong to protons on *sp^2^*-hybridized carbons with chemical shifts corresponding tannin-type polyphenols (see Appendix A) described by I. Agata et al. [34].

## 3. Materials and Methods

### 3.1. Chemicals, Samples and Extraction Protocol

The deuterated methanol, deuterated water, KH_2_PO_4_ and deuterated chloroform were purchased from Sigma-Aldrich (Saint Louis, MO, USA). Plant material was collected during the May and June 2022. on several locations in Serbia. The samples from *E. segueiriana, Neck. Subsp. seguieriana* (45°00′00.0″ N 21°01′11.5″ E), *E. panonica* (44°59′56.1″ N 21°01′09.5″ E and 44°55′48.29″ N 21°11′51.68″ E), and *E. cyparissias* L. (44°59′07.0″ N 21°01′20.45″ E) were collected at Deliblatska peščara. The samples of *E. amygdaloides* were collected at Avala (44°41′11.4″ N 20°30′53.2″ E), *E. maculata* was collected at Zemunski Kej (44°50′41.6″ N 20°24′58.5″ E) and *E. salicifolia* was collected at Košutnjak (44°45′48.6″ N 20°26′17.3″ E). The identification of plants was carried out by Marijan Niketić (Natural History Museuma, Belgrade, Serbia) and the voucher specimens were deposited at the Herbarium of the Botanical Garden “Jevremovac” University of Belgrade, Belgrade, Serbia (voucher numbers: *Euphorbia segueiriana* (BEOU17883), *Euphorbia panonica* (BEOU17884, BEOU17885), *Euphorbia cyparissias* (BEOU17893), *Euphorbia amygdaloides* (BEOU17894), *Euphorbia maculata* (BEOU17881) and *Euphorbia salicifolia* (BEOU17882)). From each species, ten samples from different locations containing aerial parts of plants were dried and stored on silica gel separately after collection. The samples were ground using a laboratory mill (IKA^®^ A11, IKA^®^-Werke GmbH & Co. KG, Staufen, Germany) frozen by the addition of liquid nitrogen and stored at −20 °C until analysis. Each sample (50 mg) was measured in 2 mL microtubes and extracted with 0.7 mL of solvent or solvents mixtures on an ultrasonic bath (BANDELIN SONOREX); then, they were centrifuged on 13,400 rpm (MiniSpin Eppendorf) and 500 µL was transferred into 5 mm NMR tubes.

### 3.2. NMR Measurements and Multivariate Data Analysis

NMR spectra were measured on a Bruker 500 AVANCE III NMR, Fällanden, Switzerland, system equipped with a 5 mm BBI probe head and BVT unit at 298K. All spectra were measured using noesypr1d pulse sequence with 16 scans for optimization and 64 scans for metabolomic studies. For each sample, the transmitter frequency was optimized, and for recorded spectra, the phase and baseline were corrected manually using TopSpin 3.6.pl7 Bruker Biospin, Rheinstetten, Germany software.

The phased ^1^H-NMR spectra were further processed using the online tool NMRProcFlow v1.4.16, INRA UMR 1332 BFP, Bordeaux Metabolomics Facility, France (https://nmrprocflow.org, accessed on 20 October 2020) for ppm calibration, global baseline correction, and local alignment. To avoid signals of the residual water, MeOH-d_4_, and CDCl_3_, respectively, the regions of δ_H_ 1.24–1.37, 3.33–3.36, and 7.24–7.26 were removed from the study. The Intelligent Binning approach (resolution factor 0.6, SNR > 10) was used to divide each spectrum into variable size buckets. To build the dataset matrix, the data were normalized to overall spectrum intensity using NMRProcFlow. SIMCA software (version 17, Sartorius Stedim Biotech, Goettingen, Germany) was then utilized for the multivariate data analysis.

## 4. Conclusions

Presented data confirmed that NMR-based metabolomics involving molecules with a wide range of polarities can be used for the chemotaxonomy of genus *Euphorbia*. The differences between the analyzed *Euphorbia* species were confirmed using SIMCA models and distance measurements between each class model. All of the species-specific samples in the prediction dataset were properly identified, and we confirmed model sensitivity and specificity by 100%.

The most important step was the optimization of extraction protocol for plants metabolomics on *Euphorbia* species. Furthermore, it was demonstrated that the presented optimization has great potential in the more efficient extraction of special metabolites characteristic for this genus and biomarkers for chemotaxonomic classification [29].

As a result of this study, a potentially new metabolite acacetin-7-*O*-glycoside was detected in *E salicifolia*, and its presence was confirmed with 2D NMR data. The variables characteristic for myrsinol diterpenes were recognized as biomarkers for *E. seguieriana*, those characteristic for jatrophanes were recognized as biomarkers for *E. amigdaloides*, and tiglianes bearing benzoate esters as biomarkers for *E. panonnica*. The triterpenes with a cyclopropane ring were identified as biomarkers for *E. cyparissias*, and in *E. maculata*, these were triterpenes and tannin-type polyphenols.

In conclusion, our experimental data conform to a previously published theoretical study [25] and confirm that our proposed protocol for NMR-based metabolomics could be used in *Euphorbia* chemotaxonomy.

## Figures and Tables

**Figure 1 plants-12-00262-f001:**
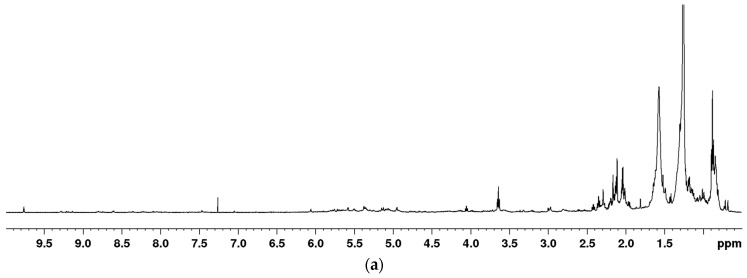
One-dimensional (1D) NOESY NMR spectra of CDCl_3_ (**a**), MEOD/CDCl_3_ (**b**), MEOD (**c**) and MEOD/D_2_O (**d**) extracts of *E. salicifolia*.

**Figure 2 plants-12-00262-f002:**
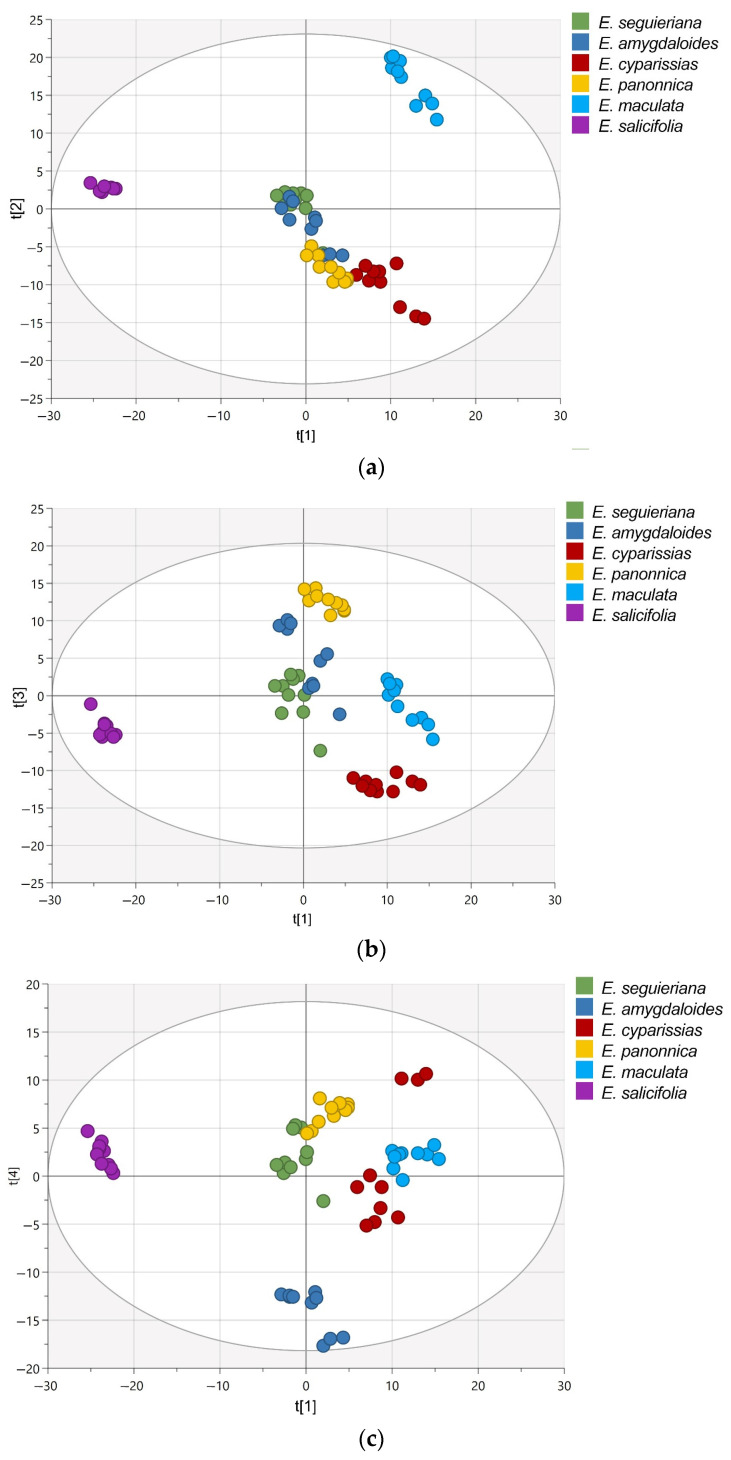
PCA score plots: PC1/PC2 (**a**), PC1/PC3 (**b**), and PC1/PC4 (**c**). Scores are colored in accordance with *Euphorbia* species.

**Figure 3 plants-12-00262-f003:**
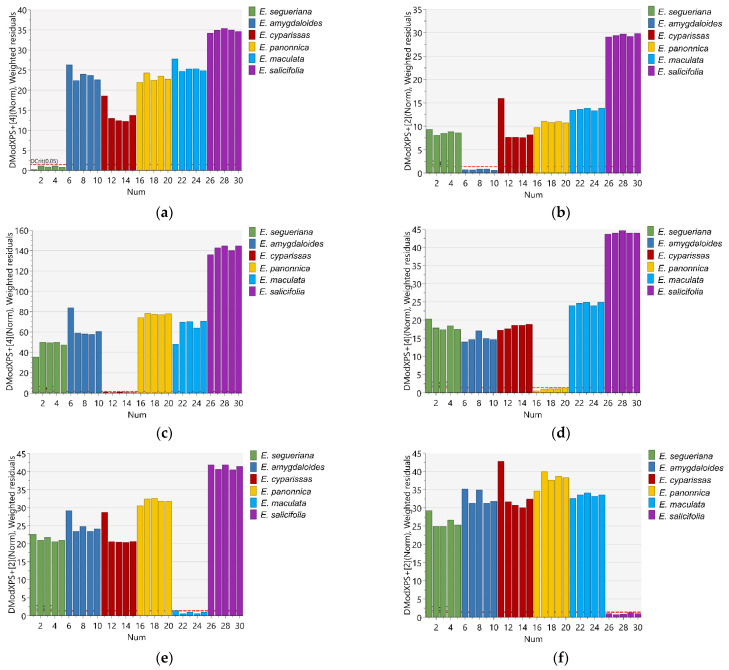
Logarithmic averaged distance to the models of (**a**) *E. segueriana,* (**b**) *E. amygdaloides,* (**c**) *E. cyparissias,* (**d**) *E. panonnica,* (**e**) *E. maculata* and (**f**) *E. salicifolia* vs. remaining *Euphorbia* species.

**Figure 4 plants-12-00262-f004:**
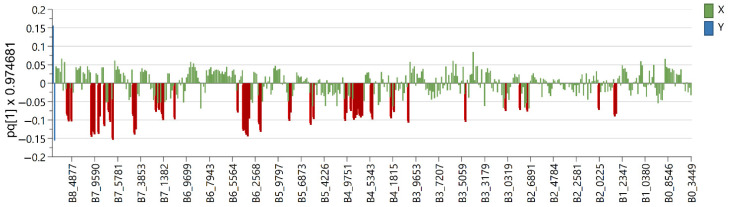
Loading plot with red and brown marked variables characteristic for *E. seguieriana* (Y variables—loading vectors scaled to unit length) in comparison to other species (X variables—chemical shift), obtained from the OPLS-DA model.

**Figure 5 plants-12-00262-f005:**
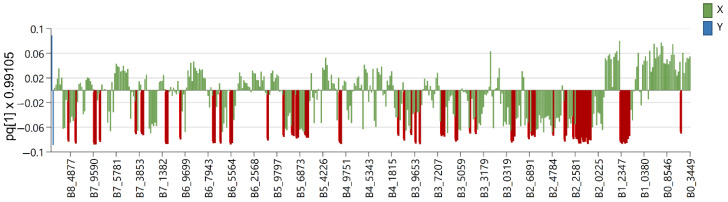
Loading plot with red and brown marked variables characteristic for *E. salicifolia* (Y variables—loading vectors scaled to unit length) in comparison to other species (X variables—chemical shift), obtained from the OPLS-DA model.

**Figure 6 plants-12-00262-f006:**
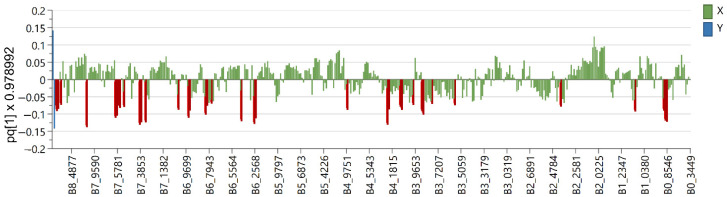
Loading plot with red and brown marked variables characteristic for *E. amygdaloides* (Y variables—loading vectors scaled to unit length) in comparison to other species (X variables—chemical shift), obtained from the OPLS-DA model.

**Figure 7 plants-12-00262-f007:**
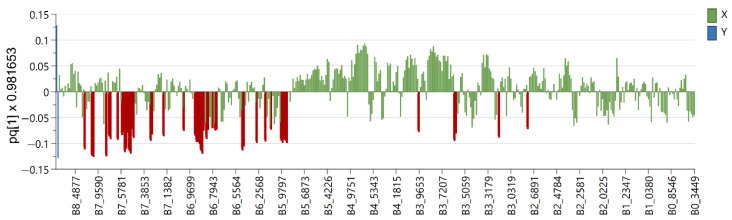
Loading plot with red and brown marked variables characteristic for *E. panonnica* (Y variables—loading vectors scaled to unit length) in comparison to other species (X variables—chemical shift), obtained from the OPLS-DA model.

**Figure 8 plants-12-00262-f008:**
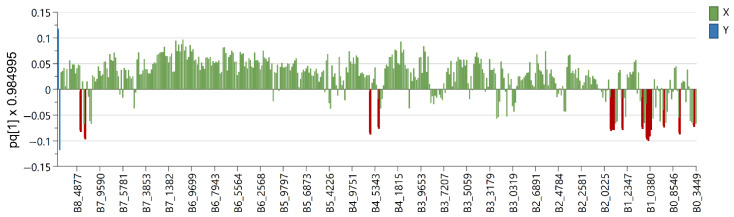
Loading plots with red and brown marked variables characteristic for *E. cyparissias* (Y variables—loading vectors scaled to unit length) in comparison to other species (X variables—chemical shift), obtained from the OPLS-DA model.

**Figure 9 plants-12-00262-f009:**
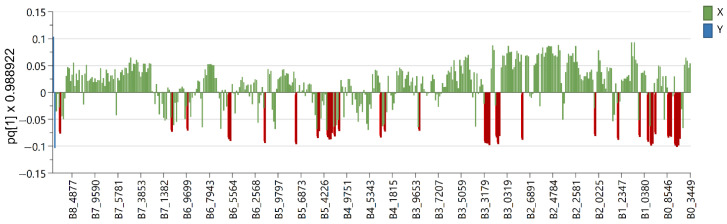
Loading plot with red and brown marked variables characteristic for *E. maculata* (Y variables—loading vectors scaled to unit length) in comparison to other species (X variables—chemical shift), obtained from the OPLS-DA model.

**Table 1 plants-12-00262-t001:** Structure, ^1^H and ^13^C-NMR spectral data of acacetin-7-*O*-glycoside.

Structure	C/H	δ_C_	δ_H_ (J in Hz)
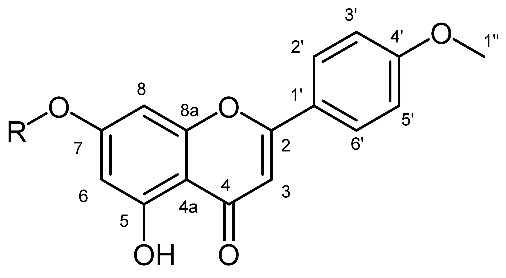	2	165.1	
	3	104.3	6.6 s
	4	182.8	
	4a	106.1	
	5	161.8	
	6	100.6	6.55 d(2.1)
	7	163.1	
	8	96.0	6.74 d(2.1)
	8a	157.6	
	1′	123.0	
	2′/6′	115.2	7.00 d(9.1)
	3′/5′	128.9	7.93 d(9.1)
	1″	55.6	3.91
	anomeric from sugars moiety	101.0	5.01 d(7.8)

## Data Availability

Not applicable.

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
