# Peer review of "Metabolomics as a Potential Chemotaxonomical Tool: Application on the Selected Euphorbia Species Growing Wild in Serbia"

_plants, 2023, doi:10.3390/plants12020262_

Round 1
Reviewer 1 Report
The authors of this manuscript employed 1H NMR for metabolomic studies on Euphorbia species. The experiments seem to be well conducted and the results scientifically sound. Nevertheless, identification of single compounds seems to be difficult with this approach, therefor a combination of (often tedious) preparative chromatographic techniques including metabolomic analyses by 1H NMR would lead to a much better dataset.
What I miss in this manuscript is a discussion why Euphorbia salicifolia appears quite different to the other investigated species. Which compounds/classes of compounds are responsible for this divergence? Was this also already detected by genetic methods by other research groups doing phylogeny of Euphorbia?
Regarding the dentatively identified acacetin-7-O-apigenin glycoside: Which sugar moiety is attached on the core structure?
One of the major points is the missing information about the samples. How many accessions from each species were extracted and analyzed? How many times was the extraction process carried out? Was the extraction exhaustively? Which parts of the plants were analyzed? Leaves or aerial parts? The presence of milk sap may influence the results, therefor only identical plant parts should be compared?
The spectra presented in figure 1 are overlapping and not very helpful because the intensity of the signals is not clearly visible. This should be improved.
Figures 4–-9: what is given on the x-axis? Probably it might be clearer when all these plots are present in a single figure.
Introduction: The correct abbreviation of circa is ca.
Please correct the typos present in this manuscript e.g. line 135 (E. salicifolia instead of E. salicipolia), line 158 E. maculata instead of E. maculate, line 174: diterpenes instead of diretpenes, line 263: measurements, line 267, frequency, line 258: frozen, line 290: confirm instead of conform. Line 158: Euphorbia species and not E. species, line 325: Euphorbia instead of Euphofrbia. Please check the whole manuscript carefully, I did not check all typos.
Line 179: the reference is not complete. In general, given names should not be used within the text. Please check the whole text.
Whole manuscript: Names of genera and species should be written in italics, check especially section 3.1 and the references. In general, the full plant names should be given after they are mentioned the first time in the text.
Check reference 30 in the reference section.
Author Response
Dear Reviewer,
We appreciate your comments.
Below are our answers to each of them individually:
- The authors of this manuscript employed 1H NMR for metabolomic studies on Euphorbia species. The experiments seem to be well conducted and the results scientifically sound. Nevertheless, identification of single compounds seems to be difficult with this approach, therefor a combination of (often tedious) preparative chromatographic techniques including metabolomic analyses by 1H NMR would lead to a much better dataset.
Response 1: The presented results are the preliminary part of the ongoing project. Our intention was not to identify individual components completely(this is planned for the next step of this research), but to optimize the extraction protocol for this challenging plant material and demonstrate the potential of NMR-based plant metabolomics.
- What I miss in this manuscript is a discussion why Euphorbia salicifolia appears quite different to the other investigated species. Which compounds/classes of compounds are responsible for this divergence? Was this also already detected by genetic methods by other research groups doing phylogeny of Euphorbia?
Response 2: The difference between E. salicifolia and other Euphorbia species arises from a different class of detected compounds. From the best of our knowledge this is the first time a flavonoid has been reported in E. salicifolia. This could be a consequence of a different approach (system of solvents) applied during extraction than those described so far in the literature for the given species, cited in the text (ref. https://doi.org/10.1016/S0040-4039(01)01285-0 and https://doi.org/10.1007/s11101-020-09667-8 )
- Regarding the dentatively identified acacetin-7-O-apigenin glycoside: Which sugar moiety is attached on the core structure?
Response 3. In order to unequivocally determine the identity of the sugar component, it’s necessary to isolate and fully characterize this compound using other instrumental techniques in addition to NMR. This will be the subject of our further research. At this moment, based on the applied methodology, we can only claim that it is acacetin-7-O glycoside. It was a typo in Abstract concerning the name of this compound.
- One of the major points is the missing information about the samples. How many accessions from each species were extracted and analyzed? How many times was the extraction process carried out? Was the extraction exhaustively? Which parts of the plants were analyzed? Leaves or aerial parts? The presence of milk sap may influence the results, therefor only identical plant parts should be compared?
Response 4. In the section 2.1. lines 113 and 114 we explained which parts of the plant we used (i. e. stems, leaves and flowers) as well as overall protocol for samples preparation (section 3.1. lines from 256 and 260). In order to further explain it, the corresponding text has been now inserted into the manuscript.
- The spectra presented in figure 1 are overlapping and not very helpful because the intensity of the signals is not clearly visible. This should be improved.
Response 5. Appropriate corrections have been made.
- Figures 4–-9: what is given on the x-axis? Probably it might be clearer when all these plots are present in a single figure.
Response 6. Appropriate corrections in Figure titles have been made. We decided to keep the individual plots. The presentation of all plots in a single figure would poduce overcrowded image, not so easy to be followed in combination with the accompanying text.
- Introduction: The correct abbreviation of circa is ca.
Response 7. Appropriate corrections have been made.
- Please correct the typos present in this manuscript e.g. line 135 (E. salicifolia instead of E. salicipolia), line 158 E. maculata instead of E. maculate, line 174: diterpenes instead of diretpenes, line 263: measurements, line 267, frequency, line 258: frozen, line 290: confirm instead of conform. Line 158: Euphorbia species and not E. species, line 325: Euphorbia instead of Euphofrbia. Please check the whole manuscript carefully, I did not check all typos.
Response 8. Appropriate corrections have been made.
- Line 179: the reference is not complete. In general, given names should not be used within the text. Please check the whole text.
Response 9. Appropriate corrections have been made and references were updated.
- Whole manuscript: Names of genera and species should be written in italics, check especially section 3.1 and the references. In general, the full plant names should be given after they are mentioned the first time in the text.
Response 10. Appropriate corrections have been made.
- Check reference 30 in the reference section.
Response 11. Appropriate corrections have been made.

Reviewer 2 Report
The authors implemented NMR-based metabolomics investigating the application of metabolomics as a state-of-the-art tool for the chemotaxonomy of plant species. The paper is very well-written and original, while the outcomes could be very helpful for discovering plant biomarkers. The paper should be published without any significant modifications. I have just a few minor comments
1) The sample size in each group (species) is relatively small for metabolomics studies in order to provide validated biomarkers. Are they the preliminary results of an ongoing project? Are there any other studies with such a small sample size?
2) A table with all the NMR-identified compounds and their peak ppm should be added by the authors.
3) The authors should add, if possible, more recent (of the last years) as references.
Author Response
Dear Reviewer,
We appreciate your comments.
Below are our answers to each of them individually
Response to Reviewer 2 Comments
- The sample size in each group (species) is relatively small for metabolomics studies in order to provide validated biomarkers. Are they the preliminary results of an ongoing project? Are there any other studies with such a small sample size?
Response 1: The presented results are a preliminary part of the ongoing project. We are aware that this is not an optimal number of samples for a genuine chemotaxonomic study and that was not the main goal here although a similar number of samples were used by Seham S. El-Hawary et al. (ref No 14.) Here we wanted to optimize the preparation procedure of this challenging plant material and show the potential of the NMR-based metabolomics. Newertheless, the validity of the presented results was confirmed through a premutation test and a cross-validation test shown in the supplementary material.
2) A table with all the NMR-identified compounds and their peak ppm should be added by the authors.
Response 2. Appropriate corrections have been made for acacetin-7-O glycoside.
We identified the spectral ranges characteristic for different classes of compounds present in the samples ( eg. myrsinol diterpenes type skeleton, specific jatrophane diterpenes named amygdaloidins, tigliane type skeletons, triterpene skeletons etc.) according to the similarity of the 1HNMR data (interpreted using 2D NMR) with those in the literature. This is presented in the text, so we thaught that it would be excessive to tabulate these data.
3) The authors should add, if possible, more recent (of the last years) as references.
Response 3. Appropriate corrections have been made and the latest references have been added.
